



# Isotopic reconnaissance of urban water supply system dynamics

Yusuf Jameel[1*], Simon Brewer[2], Richard P Fiorella[1], Brett J Tipple[3, 4], Shazelle Terry[5], and Gabriel J Bowen[1, 3]

[1]Geology and Geophysics, University of Utah, 115 S 1460 E, Salt Lake City, Utah 84112, USA.

[2]Department of Geography, University of Utah, 332 S 1400 E, Salt Lake City, Utah 84112, USA.

[3]Global Change and Sustainability Centre, University of Utah, 115 S 1460 E, Salt Lake City, Utah 84112, USA.

[4]Department of Biology, University of Utah, 257 S 1400 E, Salt Lake City, Utah 84112, USA.

[5]Jordan Valley Water Conservancy District, 8215 S 1300 W, West Jordan, Utah 84088, USA.

*Correspondence to: yusuf.jameel@utah.edu

**Abstract.** Public water supply systems (PWSS) are critical infrastructure that are vulnerable to

contamination and physical disruption. Exploring susceptibility of PWSS to such perturbations

requires detailed knowledge of supply system structure and operation. The physical structure of

the distribution system (i.e., pipeline connections) and basic information on sources are

documented for most industrialized metropolises. Yet, most information on PWSS function

comes from hydrodynamic models that are seldom validated using observational data. In

developing regions, the issue may be exasperated as information regarding the physical structure

of the PWSS may be incorrect, incomplete, undocumented, or difficult to obtain in many cities.

Here, we present a novel application of stable isotopes in water (SIW) to quantify the

contribution of different water sources, identify "static" and "dynamic" regions (e.g., regions

supplied chiefly by one source vs. those experiencing active mixing between multiple sources),

and reconstruct basic flow patterns in a large, complex PWSS. Our analysis, based on a Bayesian

mixing model framework, uses basic information on the SIW and production volumes of sources

but requires no information on pipeline connections in the system. Our work highlights the

ability of SIW to analyze PWSS and document aspects of supply system structure and operation



that can otherwise be challenging to observe. This method could allow water managers to

document spatiotemporal variation in flow patterns within PWSS, validate hydrodynamic model

results, track pathways of contaminant propagation, optimize water supply operation, and help

monitor and enforce water rights.

## 1. Introduction

The world is becoming increasingly water stressed due to growing population and the

intensification of agricultural and industrial activities (*Arnell*, 1999; *Vörösmarty et al.*, 2010;

*Arnell and Lloyd-Hughes*, 2014; *Haddeland et al.*, 2014; *Hejazi et al.*, 2015). Water managers

have resorted to overexploitation of groundwater (*Rodell et al.*, 2009; *Wada et al.*, 2010; *Gleeson

et al.*, 2012), large-scale inter-basin transfers (*Davies et al.*, 1992; *Meador*, 1992; *Ghassemi and

*White*, 2007), desalinization of seawater (*Khawaji et al.*, 2008; *Elimelech and Phillip*, 2011), and

recycling of wastewater (*Yi et al.*, 2011) to meet these increasing water demands. Water

produced from these sources must often be transported through a long and complicated network

of distribution lines to provide safe and clean potable water at the point-of-use. The complexity

of public water supply systems (PWSS) can vary widely, ranging from linear, single-source

distribution systems to branched distribution networks using multiple water sources and complex

storage systems. Regardless of structure, these systems are critical infrastructure that are

vulnerable to a wide range of potential threats including supply contamination and infrastructure

failure to climate change. To understand the stability of water supplies, conduct risk evaluation,

and develop effective and efficient responses for particular threats, it is critical to understand the

physical and spatial structure of the distribution network, connectivity within the system, and the

links between the point-of-use and environmental water sources.

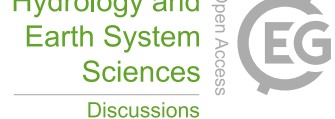

The physical structure of the distribution system and basic information on water sources are generally well documented for many first-world metropolises. In these settings, water managers traditionally rely on network analyses to study different aspects of water distribution systems,

including pressure gradients, flow rates, water losses from the supply system, identification of vulnerable sections, and tracking of disinfectants and contaminants (*Boryczko and Tchórzewska-Cieślak*, 2014; *Pietrucha-Urbanik*, 2015; *Yoo et al.*, 2015). These analyses are generally robust; however, they are seldom validated using observational data and can suffer from shortcomings including the absence of unique solutions in underdetermined systems, assumption of invariant

flow rates, uncomprehensive or non-inclusiveness of uncertainty in the analysis (*Waldrip et al.*, 2016), and outdated/incorrect information on infrastructure (*Liggett and Chen*, 1994). Beyond statistical and computational issues, hydrodynamic modelling requires extensive and detailed information about the PWSS, including node elevation, pipe length and diameter, and pump operating data. For many cities in the developing world, where distribution networks are

commonly unregulated and decentralized, even basic information on supply system structure and source contributions may be incorrect, incomplete, undocumented, or difficult to obtain. Hydrodynamic modeling of PWSS in such cases can be challenging and prone to significant errors.

It is important to develop techniques that can be applied to study PWSS with minimal

information on the physical structure and connectivity within the supply system, given the growing water security challenges due to climate change (*Arnell*, 1999; *Vörösmarty et al.*, 2010), expanding complexity and dynamicity of urban water systems, and increasing detrimental effects of aging water infrastructure in many countries (*Hanna-Attisha et al.*, 2016; *Kaushal*, 2016; *Larsen et al.*, 2016; *Schnoor*, 2016). Such methods will not only provide observational validation



to hydrodynamic models in first-world cities but will also help in developing understanding of

interactions of water sources, supply dynamics and water quality analysis within the distribution

system that can then be applied to cities in the developing world. In this regard, new

computational techniques are being developed to understand failure in the water distribution

system with imprecise, limited and ambiguous information on the supply structure (*Najjaran et*

*al.*, 2005; *Ismail et al.*, 2011; *Bolar et al.*, 2013; *Kabir et al.*, 2015) or analyze the water

distribution system in a probabilistic framework (*Waldrip et al.*, 2016). Here, geochemical

tracers such as the stable isotopes in water (SIW) can also serve as a tool to study water

management within complex urban distribution systems. Recent studies have shown that

distributions of the SIW in urban areas relying on multiple water sources can be used to

characterize active water management practices, identify linkages between socioeconomic

factors and water management practices, and quantify the effects of climate variability on water

resources (*Jameel et al.*, 2016; *Tipple et al.*, 2017).

Stable isotopes in water are natural and conservative tracers documenting provenance

information and have been used extensively in climatological (*Rozanski et al.*, 1992; *Gat*, 1995;

*von Grafenstein et al.*, 1999; *Aggarwal et al.*, 2005; *Dutton et al.*, 2005), ecological (*Hobson*,

1999; *Hobson et al.*, 1999; *Bowen et al.*, 2005a; *Wassenaar et al.*, 2009), ecohydrological

(*Dawson and Ehleringer*, 1991; *Jasechko et al.*, 2013; *Evaristo et al.*, 2015; *Evaristo and*

*McDonnell*, 2017; *Matheny et al.*, 2017) , forensic (*Bowen et al.*, 2005a; *Bowen et al.*, 2005b;

*Bowen et al.*, 2007; *Kennedy et al.*, 2011; *Landwehr et al.*, 2014; *Ueda and Bell*, 2017), and

hydrological studies (*McDonnell et al.*, 1991; *Klaus and McDonnell*, 2013; *Good et al.*, 2014;

*Gorski et al.*, 2015; *Jefferson et al.*, 2015; *Gabor et al.*, 2017; *Jameel et al.*, 2018). Within the

terrestrial hydrological cycle, significant isotopic differences between water sources (i.e., river,

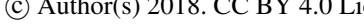

lakes, reservoirs, shallow and deep groundwater, or recycled water) can exist at catchment, regional, and global scales due to seasonal biases in recharge, differences in meteoric water

composition, altitude, and meteorological factors such as temperature, humidity and wind speed. Within the natural realm, these differences have been exploited to understand biogeochemical and hydrological processes and trace and partition sources and contaminants (see references above). However, the application of SIW in human-managed water systems in general, and specifically in the context of understanding the dynamics of water distribution systems, has been

limited.

Here, we present an isotopic survey of the Jordan Valley Water Conservancy District (JVD) service area within Salt Lake Valley metropolitan area (SLV) of northern Utah, USA (Fig. 1), which is a multi-source water distribution network  and attempt to understand mixing between water sources at various sites (subsequently referred to as distribution sites) distributed on the

transmission lines. During our survey (May 2015 – October 2015), most of the distribution sites cluster along a major source that appear to be consistent with a single source; however, few sites did not cluster along any of the major sources, suggesting water obtained at these sites is delivered from multiple sources (Fig. 2) . Using information on the production volume from the different sources, we analyze the stable isotope data at a monthly resolution within a Bayesian

framework to generate quantitative estimates (with uncertainty) of the contribution of individual sources at the distribution sites. These analyses reveal basic information on supply and transport dynamics within the system, reflecting the physical structure of the supply system and the geographic distribution of sources. Finally, we combine the monthly analyses to characterize the spatial structure of the system in terms of contribution areas for the different sources across the

supply network. Our results suggest that SIW-based Bayesian isotope mixing models (BIMM)



could be a powerful and useful tool to interrogate PWSS, providing observational validation to hydrodynamic models, tracking contaminants and disinfectants within the supply system, and providing a tool for monitoring and enforcing of water rights in PWSS managed by or for multiple stakeholders.

## 2. Methods

### 2.1 Site description

The JVD is a wholesale supplier to 17 water districts in the Salt Lake Valley and retails directly to several locations in SLV located primarily on the northeastern part of the valley (known as Jordan Valley retail area, Fig. 1). As a wholesaler, JVD sells water to these 17 districts from fixed locations on the JVD distribution line and is not responsible for managing and distributing water in these districts beyond the transfer point.

In general, JVD relies on 3-5 sources at any given time to supply water to its service area; however, during the summer season (June – August) an additional 5-7 sources are often used to meet increased water demand (personal communication, JVD operations manager). Water is sourced primarily through the Provo River system (>75% of total water supplied annually), and is supplemented with water from Wasatch creeks and groundwater wells depending on demand. The Wasatch creek sources carry runoff from snowmelt in the Wasatch Mountains (Fig. 1) and are used only in spring and early summer. There are approximately 25 active groundwater wells managed by the JVD. Not all wells operate simultaneously, rather only 2-5 wells operate at any given time and the operating wells are rotated every few months.



JVD operates three water treatment plants (WTP). The Jordan Valley Water Treatment Plant (JVTP) is the largest water treatment plant and is situated at the southern end of the valley. It has a maximum operational capacity of 180 million gallons per day (MGD, 681374 m$^3$ per day) and treats water from the Provo River. The South East Water Treatment Plant (SETP) is a significantly smaller WTP (maximum operational capacity of 20 MGD/ 75708 m$^3$ per day) situated on the southeastern side of the valley. It also treats water from the Provo River, but during spring and early summer (Mid-April to June) most of the water treated at SETP is from the Wasatch creeks. The South West Water Treatment Plant (SWTP, maximum operational capacity of 7 MGD/ 26497 m$^3$ per day) is located in the middle of the valley and treats water from groundwater wells located near the treatment plant. Groundwater wells supplying the SWTP (shown as dark blue squares in fig. 1) have a high salt concentration and require extensive purification before being pumped into the distribution system. In contrast, groundwater wells located on the eastern side of the valley (shown as light blue squares in fig. 1) have lower concentrations of dissolved salts and do not require additional treatment before entering the distribution system (personal communication, JVD operations manager).

The JVD water distribution system consists of one primary (Fig. 1), several secondary (line 2 through 6, Fig. 1) and numerous tertiary transmission lines. Water can move in either direction in all the transmission lines, however in transmission line 1, water primarily moves from south to north. Water from JVTP is pumped directly into transmission line 1. SETP water is pumped into transmission lines 2 and 3 (Fig. 1). Water from SWTP is supplied mainly to residential areas in the vicinity of the WTP (these supply connections are not shown in fig. 1), though some water from SWTP is also pumped directly into transmission lines 5 and 6 (bypassing line 1). Water from wells in the eastern side of the valley is pumped directly into the transmission lines on

which the respective wells are located. Most of the secondary transmission lines are interconnected via tertiary and quaternary lines (not shown in Fig. 1 except for the tertiary connections in the Jordan Valley retail area).

## 2.2 Sample acquisition and processing

Each month from May to October 2015, we sampled water at sources contributing to the JVD service area and at numerous locations ("distribution sites" or simply as "sites", Figure 1) on the

JVD transmission lines. Source water samples were collected as effluent from the WTPs and directly from the groundwater wells, while distribution site samples were collected from monitoring taps on the transmission lines. The distribution sites are routinely monitored by JVD for water quality analysis and are located across the supply network based on JVD's monitoring program. As such, the distribution sites are more densely distributed in the Jordan Valley retail

area because JVD is responsible for water quality monitoring within this area. In other districts, where JVD wholesales water, samples were collected only from the primary and secondary transmission lines. Samples were collected in 4-ml clean glass vials and stored in a refrigerated at 4°C prior to analysis.

Sources and distribution sites were sampled 1-3 times per month. Surface water sources (Provo

River and Wasatch creeks) were sampled each month; however, some of the wells were not sampled in their month of operation. In these cases, the values observed at the same well during other months were used to characterize water supplied from this well. This substitution was justified given that previous work showed little temporal variability in the isotope values of water supplied from SLV groundwater wells (*Jameel et al.*, 2016).





SETP-treated water was sourced mostly from the Wasatch creeks in May and June 2015, and

from Provo River from July to October 2015. JVTP-treated water was sourced from Provo River

for the entire period of analysis. Therefore, we considered SETP and JVTP as separate sources in

May and June and as a single source from July to October. Isotope ratios for effluent from SETP

and JVTP were not statistically different between July and October (Hotteling multivariate t-test,

p > 0.05). Additionally, groundwater wells situated close to each other and having similar

isotope values (differences in for $\delta^2H$ and $\delta^{18}O$ less than 0.5‰ and 0.1‰ respectively) were also

combined together for our analyses (such as wells 64S and 70S in June, July and August, 2015).

### 2.3 Isotope analysis

The samples were analyzed within few weeks of collection at the Stable Isotope Ratio Facility

for Environmental Research (SIRFER) facility, University of Utah, on a cavity ring-down

spectrometer (L2130-i; Picarro, Inc., Santa Clara, CA) following protocols described in (*Good et

al.*, 2014), after (*Geldern and Barth*, 2012). Values are reported in $\delta$ notation: $\delta = (R_{sample} /$

$R_{standard} - 1)$, where $R_{sample}$ and $R_{standard}$ are the $^2H/ \ ^1H$ or $^{18}O/^{16}O$ ratios for the sample and

standard, respectively, and the VSMOW standard is referenced (*Coplen*, 1988). Accuracy and

precision were checked using a secondary laboratory reference water, and the analytical

precision for these analyses were ±0.3‰ for $\delta^2H$ and ±0.03‰ for $\delta^{18}O$ (± 1 SD).

### 2.4 Bayesian mixing model and statistical analyses

We estimated the fractional contribution of the different sources at the distribution sites for each

month using a Bayesian Isotope Mixing Model (BIMM). The advantages of a Bayesian approach

include: (1) simultaneous analysis of both isotope ratios ($\delta^2H$ and $\delta^{18}O$), (2) inclusion of prior

information into the statistical analysis, (3) explicit incorporation of analytical and sampling



uncertainties into the model, and (4) robust estimates of uncertainty and quantification of most likely solutions in an underdetermined system (number of sources greater than number of isotopes plus one).

The Bayesian mixing model described here is similar to those used in other studies involving stable isotope data (*Ogle and Barber*, 2008; *Parnell et al.*, 2010; *Cable et al.*, 2011; *Mailloux et al.*, 2014). For our analysis, we first define the likelihood of the source isotope data. For this, we assumed that the different isotopic observations of each source *(J)* for a given month are coming from a bivariate normal distribution with a mean vector $[\mu\delta^2H_J , \mu\delta^{18}O_J]$ and a precision

matrix($\Omega_J$, inverse of a covariance matrix) that reflects the temporal variability in the source isotope values. Thus,

$$\begin{bmatrix} \delta^2H_{1J} & \delta^{18}O_{1J} \\ \vdots & \vdots \\ \vdots & \vdots \\ \delta^2H_{NJ} & \delta^{18}O_{NJ} \end{bmatrix} \sim Mnormal\left(\left[\mu\delta^2H_J , \mu\delta^{18}O_J\right], \Omega_J\right), \tag{1}$$

where $\delta^2H_{1J} \ldots \ldots \delta^2H_{NJ}$ and $\delta^{18}O_{1J} \ldots \ldots \delta^{18}O_{NJ}$ are the N observations of $\delta^2H$ and $\delta^{18}O$ of source $J$ for that month, $[\mu\delta^2H_J , \mu\delta^{18}O_J]$ is the mean vector and $\Omega_J$ is the precision matrix.

Similar to the source model, we assumed that for a supply site *(I)*, the monthly averaged isotope values $[\delta^2H_I , \delta^{18}O_I]$ follow a bivariate normal distribution with mean vector $[\mu\delta^2H_I , \mu\delta^{18}O_I]$ and a precision matrix ($\Omega_I$). Thus, for a supply site *(I)*:

$$[\delta^2H_I , \delta^{18}O_I] \sim Mnormal\left(\left[\mu\delta^2H_I , \mu\delta^{18}O_I\right], \Omega_I\right) \tag{2}$$

The mean stable isotope values of the supply site can also be expressed as a mixing model,

where the mean value for supply site $I$ ($\mu\delta^2H_I , \mu\delta^{18}O_I$) is the sum of the mean values of the





sources weighted by their fractional contributions. Therefore, if $K$ sources were used in a given month, $(\mu\delta^2 H_I , \mu\delta^{18} O_I)$ for each supply site ($I$) is denoted by:

$$[\mu\delta^2 H_I , \delta\mu^{18} O_I] = \sum_{J=1}^{J=K} (f_J) \; [\mu\delta^2 H_J , \delta\mu^{18} O_J] \qquad (3)$$

where $f_J$ is the proportional contribution from a given source $J$ at supply site $I$. Values of $f$ were

described using the Dirichlet distribution, a multivariate generalization of the beta distribution that follows the mass-balance constraint i.e. $0 \le f_J \le 1$ and $\sum_{J=1}^{J=K}(f_J) = 1$. The Dirichlet distribution is characterized by parameter vector $\alpha = \{\alpha_1, \ \alpha_2, \ \alpha_3, ..., \alpha_K\}$, such that the mean value associated with each $f$ is $f_J = \alpha_J / \sum\{\alpha_1, \ \alpha_2, \ \alpha_3, ..., \alpha_K\}$.

The default non-informative prior assigned to the Dirichlet distribution is the Jeffreys prior,

where each element of the vector $\alpha$ is assigned a value of $1/K$ (with $K$ being the number of sources) or a value of 1assigned to each element of $\alpha$ (*Parnell et al.*, 2010). However, more informative prior distributions can also be used. We assigned prior values for each supply site based on the relative volume of water supplied by each source and their Euclidian distance from the respective distribution sites. First, we assumed that the probability of a source supplying a

given distribution site was proportional to the volume of water that source supplies to the JVD distribution system. Thus, sources contributing more water to the JVD system have a higher probability of supplying water to any given site than do lower-volume sources. Second, we assumed that the probability of a source supplying water a given site was inversely proportional to the distance between the source (e.g., water treatment plant or well location) and the

distribution site. Thus, sources closer to a distribution site have a higher probability of supplying



water at that site. We combined both pieces of prior information to obtain a normalized prior estimate, as described below.

In the first step, we calculated prior weights for the Dirichlet parameters for each source based upon the proportional volume of water produced ($V$) by that source:


$$\alpha_{J\_volume} = \frac{V_J}{\sum_{J=1}^{J=K} V_J}$$

(4)

Second, we distance-weighted each source's prior inversely based upon its distance ($D$) from supply site $I$:

$$\alpha_{JI\_distance} = \frac{\frac{1}{D_{JI}}}{\sum_{J=1}^{J=K} D_{JI}}$$

(5)

We then combined the volume and distance weighted priors to obtain a prior estimate of the

mean contribution from source $J$ at supply site $I$:

$$\alpha_{JI_{prior}} = \frac{\alpha_{J\_volume} * \alpha_{JI\_distance}}{\sum_{J=1}^{J=K} \alpha_{J\_volume} * \alpha_{JI\_distance}}$$

(6)

For example, if there were three sources supplying 3000 m$^3$, 1500 m$^3$ and 1500 m$^3$ of water to the JVD system that were located 4 km, 1 km and 10 km away from supply site $I$, then the Dirichlet prior vector would be {0.3125, 0.625, 0.0625} for this site $I$. The prior contributions of

selected sources at the distribution sites for June 2015, based upon the above-described method, in spatial and isotope space are shown in fig. 3 and fig. 4 respectively.

We estimated the posterior fractional ($f$) contributions to each site $I$ from each source using the JAGS software package (*Plummer*, 2003), which can be integrated in the R statistical language





using different R packages (*Plummer*, 2013; *Denwood*, 2016). We ran 3 parallel Markov chain

Monte Carlo (MCMC) simulations for 300000 iterations per chain, which were thinned every 50

steps. The first 40,000 iterations were discarded as burn-ins, providing us with 5200 samples for

calculating the posterior statistics. We checked the convergence using the coda package

(*Plummer et al.*, 2006) and Gelman diagnostics (*Gelman et al.*, 2014). All statistical analysis was

performed in R (*R core team*, 2018).

**2.5 Model results interpretation and cross-validation**

For qualitative interpretation and to identify spatiotemporal variations in the association between

sources and distribution sites within JVD, we considered any distribution site that our mixing

analysis suggested was receiving more than 70% (mean contribution) of its water from a single

source to be supplied predominantly by that source. Sites where the analysis suggested less than

70% water came from a single source were considered to receive water from multiple sources.

For each month, we compared the fractional production volume of each source with the fraction

of the service area that our analysis suggested was served by the source. We calculated the areal

contribution of the different sources for each month as a cross-check of the results obtained by

BIMM. As a first-order approximation, we expected strong agreement between volumetric and

areal contribution of a source as the area supplied by a source should be proportional to its

volumetric supply.  To calculate the areal coverage of a given source, we first calculated the area

of influence ($A_I$) of each site on the transmission line, defined as the area of the Thiessen polygon

associated with the site. For each source $J$, values of $A_I$ were multiplied by the mean fractional

contribution from that source ($f_{I,J}$) . The resulting values were summed across all distribution

sites ($\sum A_I X f_{I,J}$) and divided by the total area of JVD supply region.



## 3. Results and discussion

### 3.1 Sources and distribution sites isotope ratios

Source water isotope values, measured across all months ranged from -16.67‰ to -14.86 ‰ for $\delta^{18}O$ and -122.5‰ to -114.1‰ for $\delta^2H$. Four sources (JVTP, SETP, SWTP and well 64S)

operated for the entire sampling period, and other sources operated intermittently. For each month, approximately 90% (or more) of the water was supplied by the three WTPs (JVTP, SETP and SWTP), with majority being supplied by JVTP. Groundwater wells situated on the eastern side of the valley contributed approximately 10% of the total water supplied each month, with well 64S supplying 1-3% of the total volume each month.

The isotope values of SWTP and well 64S were distinct (Hotteling multivariate t-test, $p < 0.05$) from each other and from those of JVTP and SETP for all months (Fig. 2). Isotope ratios of JVTP and SETP water were distinct (Hotteling multivariate t-test, $p < 0.05$) for May and June 2015, only. From July 2015 onwards, water from the Provo River was used by both of these WTPs; therefore, similar isotope ratios were expected. Well 64S had the lowest isotope ratios

measured for any source, and exhibited high d-excess values (~10‰), where d-excess is defined as $\delta^2H - 8\delta^{18}O$ (*Dansgaard*, 1964). Values from SWTP, in contrast, showed evidence for evaporative isotope effects, with high $\delta^{18}O$ values and low d-excess (~4.2‰). JVTP isotope ratios increased from May to October, 2015, as did SETP isotope ratios from July to October, 2015, which can be due to evaporative enrichment of the heavy isotopes in upstream reservoirs

of the Provo River system from spring to fall (mean d-excess for JVTP in June 2015 was 5.19‰ and in October 2015 was 3.93‰).

The most and least negative isotope values of water from distribution sites were similar to the values observed for well 64S/70S and SETP, respectively. With the exception of a few sites in the May 2015 survey, distribution site isotope ratios fell within the convex hull defined by the

source waters (Fig. 2). For each month, a number of distribution sites exhibited values similar to JVTP (Fig. 2). Clustering of supply site values was also observed near well 64S and SETP source values. At no point during the study did we observe any distribution sites with isotope values similar to those of SWTP source water (Fig. 2). For all months except October, approximately 20-30% of the supply site values did not cluster near any major source, but rather

were situated between sources. This pattern is consistent with expectations for mixing of water from multiple sources within this PWSS.

### 3.2 Source contributions at the distribution sites

We first illustrate the implementation of the BIMM for June 2015 (Fig. 5). Our model builds upon the work of Jameel et al. (2016) and Tipple et al. (2017), but goes beyond their analyses by

providing quantitative, spatially- and temporally-resolved estimates of source contributions at locations throughout the SLV supply system.

According to our model, most of the distribution sites (45 out of 65) received most (>70%) of their water from a single source. At all of these sites, the dominant source identified was either JVTP (24 sites, Fig 4e and 5a), SETP (15 sites, Fig 4f and 5b), or well 64S/70S (6 sites, Fig. 5d).

This shows that three of the four largest sources operating at the time dominated the supplies of a large number of sites, and that the number of sites served by these sources was approximately proportional to the volumetric contribution from each source. Our analysis suggests that the

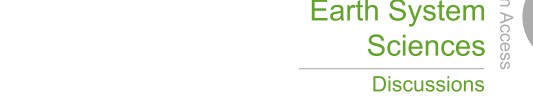

remaining 20 sites did not receive water predominantly from a single source, but had contributions from multiple sources.

Most of the sites receiving large proportional contributions from JVTP, SETP and well 64S/70S were located on the transmission lines known to be directly connected to these sources (Fig. 5a, 5b and 5d). In contrast, distribution sites distant from all sources were more likely to exhibit mixing between multiple sources. During June 2015, all but three sites in the Jordan Valley retail area showed evidence for source mixing.

Our model output, in context with known physical infrastructure (i.e., pipelines) and geographic locations of the sources, suggested patterns of source-supply connectivity within the JVD. Our results suggest: 1) subtle differences in mixing proportions among distribution sites receiving water mainly from the two largest sources (JVTP and SETP), 2) limited mixing at distribution sites located on transmission lines receiving water from multiple sources, and 3) bypassing of a

specific transmission line during water transport. Below, we discuss each of these observations in more detail.

For sites on the western portion of the JVD, the model-inferred mean JVTP contributions were uniformly large (>90%), suggesting that JVTP was likely the dominant source supplying water to these sites (Fig. 5a). This was expected, as most of these sites have limited connectivity to

other sources apart from the SWTP. In contrast, our model suggested that most distribution sites receiving water predominantly from the SETP had mean contributions of SETP waters of 70-90% (Fig. 5b). This likely reflects minor contributions of water from JVTP and several smaller sources in close proximity to SETP (Fig. 5a, 5e and 5f), implying that although these sites are served chiefly by a single source, they also receive a significant fraction of water from other





sources, and thus be exposed to any supply or contamination issues associated with those minor sources. According our model, sites receiving more than 60% of water from SETP had an average contribution of 12% from the minor sources (excluding JVTP) with some sites receiving as much as 27% of water from the minor sources.

Our model suggested limited mixing between JVTP, SETP and other minor sources for

distribution sites located on transmission lines 2 and 5 (Fig. 5a) that could receive water from all of these sources. Distribution sites on these lines mainly received water either from JVTP or SETP (more than 70%), with contributions from other sources generally less than 30% (Fig. 5a and 5b). Considering that JVTP supplied more than 80% of all water in June 2015, we expected mixing and a large contribution from the JVTP at sites along these transmission lines (lines 2 and

5). One factor that may have caused limited mixing between these sources within the supply lies is the higher elevation of SETP (1532 m) and other minor sources on the eastern side of the valley compared to JVTP (1424 m). We suggest that the higher gravitational potential energy of water introduced from SETP and minor sources may create a pressure differential that limits mixing between these two sources; however this remains a hypothesis to be tested.

Our model suggested negligible presence of SETP water in transmission line 3 (< 15%), whereas the mean contribution of SETP in a closely running parallel transmission line 4 was high (> 60%, Fig. 5b). This result implies that water moving northward from SETP bypassed line 3. This is most likely due to the presence of well 64S/70S on line 3, which our results suggested was the principal source to all the sites on line 3. This highlights the ability of isotope mixing model to

capture small-scale interactions between sources and supply connections.

The BIMM presented here is blind to the actual physical connections in the JVD service area. Nonetheless, our results closely match the specific linkages between sources and distribution sites along known transmission lines. The ability of BIMM to identify patterns of source-supply connectivity within this system suggests potential to use similar SIW-based methods to obtain

information from less documented PWSS.

### 3.3 Assessment of uncertainties and model limitations

In addition to providing point estimates of source water contributions, the BIMM also provides estimates of uncertainty. To analyze the uncertainties, we divided the isotope values of the distribution sites measured in June 2015, into three groups. Group 1 consisted of sites with

isotope values similar to one of the major sources (Fig. 6a), group 2 consisted of sites with isotope values in-between the SETP and JVTP endmembers (Fig 6b), and group 3 consisted of distribution sites with isotope values similar to one of the minor sources but significantly different from any of the major sources (Fig 6c).

According to our model, all sites in group 1 had large contributions (> 70%) from one of the

major sources (JVTP, SETP or 64S/70S), and we observed narrow 95% credible intervals (CIs, ranging mostly from 0.6-1) for the proportional contributions from these sources (Fig. 7a and 7b). At these sites the CIs for other sources were also narrow and ranged from 0.0-0.3 (Fig. 7c-h), indicating high levels of confidence that other sources were minor contributors to these sites. The effectiveness of BIMM in providing tight and robust posterior distributions for group 1 sites

is due to the strong similarity between source and distribution site isotope values in this group and the distinct isotope ratios of water from these sources relative to all others.





For group 2 sites, the model predicted mixing primarily between water from the JVTP and the SETP (Fig. 7a and 7b), with contributions of 40% to 60% from each of these sources. According to our model, the contribution from the Solena Way well was minimal at all group 2 sites except

for a single site situated very close to the well (Figure 5e). Given that the Solena Way well contributed only 1% of the total water to the system, dominance of JVTP and SETP water at most group 2 sites is reasonable. Further, most of the group 2 sites were situated in the Jordan Valley retail area, far from the Solena Way well (> 5 km, Fig. 5a, 5b and 5e). The CIs associated with different sources at these sites were larger than group 1 sites. Most of these sites had CIs

ranging from 0.0-0.6 for JVTP (Figure 7a), from 0.3-0.6 for SETP (Figure 7b), and from 0.0-0.6 for the Solena Way well (Figure 7g). The tighter credible intervals of contributions from SETP compared to JVTP and Solena Way Well at these sites suggests that our model is more confident about the contribution from SETP (i.e. between 30% and 60%) compared to contributions from JVTP and Solena Way Well. As observed for group 1, the CIs associated with other sources were

small, in general, ranging from 0.0-0.2. The advantages of including distance and volume effects in our model were reflected in this group, as our model preferred mixing between water from the JVTP and the SETP over possible mixtures between water from the Solena Way well and other minor sources.

Group 3 exhibited isotope values that were distinct from all major sources and were similar to

one or more minor sources. According to the model, no source (major and minor included) contributed more than 50% (mean) at these sites. In general, for a given supply site in this group, our model assigned the highest mean contribution to the minor source with isotope ratios most similar to those of the supply site water (for example see Fig. 5e). The CIs associated with proportional contributions from the different sources were large, however, and for some sources

ranged from 0.0-0.9 (Fig. 7h). This suggests that more than one possible source or multi-source

mixture was consistent with the isotopic and prior constraints for these sites, resulting in

identifiability issues that are commonly observed in isotope mixing models (*Cable et al.*, 2011;

*Erhardt and Bedrick*, 2013; *Parnell et al.*, 2013). In our case, non-unique assignments for group

3 sites arose due to the presence of multiple sources with comparable isotope values near the

distribution sites and also due to several probable potential mixing solutions between SETP,

64S/70S well and these minor wells. The issue was compounded further by similar and low prior

probabilities associated with the minor sources making it difficult for the model to identify one

distinct source as a major contributor.

Our results highlight the robustness as well as the limitations of our model. Both the use of

informative priors and the comprehensive assessment and interpretation of uncertainty are likely

to improve the quality of inferences drawn from our method. A key outcome of the priors

specified here is that volumetrically minor sources were not identified as a major contributor to

distribution sites, even though in many cases they had similar isotope values, except in cases

where proximity provided additional evidence suggesting that they were likely sources. This

result was also observed in July and August 2015. Consideration of credible intervals estimated

in the analysis shows substantial and interpretable variation in the confidence of source-water

estimates among different sites. Even in cases where relatively high mean source contributions

were assigned to a given site, robustness in the model solutions can be recognized through

review of credible intervals and used to more accurately interpret these results.




### 3.4 Spatiotemporal variations in source contributions

We extended our analysis to all months from May to October, 2015, to assess changes in the patterns of water distribution as water demand, source types, and production volumes changed

through the sampling period (Fig. 2).

Mixing between sources was high in May, with only 25% of the distribution sites receiving more than 70% of their water from a single source (Fig. 8a). For May, most of the distribution site values were intermediate to the source water values (Fig. 2), clearly indicating substantial mixing across most parts of the distribution system. A handful of supply site samples in May also fell

outside of the convex hull defined by the sources, suggesting that our sampling may not have captured all contributing sources, but the conclusion of pervasive mixing is not likely to be affected by this omission. In contrast, our model suggests that almost 70% of the sites were supplied chiefly by a single source in June and July, with this value increasing to more than 75% in August and September (Fig. 8b-e). By October, the supply system had transitioned to a single,

major source, and our results showed no significant mixing between sources for that month (Fig. 8f). Except for May 2015, where we observed large-scale mixing between different sources throughout JVD, distribution sites receiving water from multiple sources were limited mostly to the Jordan Valley retail area. Since this area is distant from all major sources and is surrounded by multiple transmission lines, mixing observed at the distribution sites is not surprising.

Perhaps the most surprising part of our analysis was our inability to detect contributions from SWTP at the distribution sites, even though this source supplied 3-5% of total water production each month. Small contributions (10% to 20%) from this source were indicated at couple of sites on transmission line 5, situated relatively far from SWTP, during June, July and August 2015

(see Fig. 5c for June). However, this source was not identified as the predominant source (i.e.

>70%) at any distribution sites, including those closest to SWTP, during the study. According to

JVD operations managers (personal communication) most of the water from SWTP is supplied

to a residential area in the immediate vicinity of the treatment plant, and none of our distribution

sites were located in this area. A small fraction of the SWTP water is routed to the western part

of the JVD, which is possibly reflected in our results suggesting minor contributions from this

source to distribution sites along distribution line 5.

We combined our model output for different months to highlight variability and quantify the

mean source contribution for each source at the different distribution sites from May to October

2015 (Fig. 9). Our result suggests that most of the sites received water from multiple sources or

switched sources during our analysis period with the exception of a few sites receiving Provo

River and 64S/70S well water for all the six months. Our results show significant changes

throughout the sampling period, highlighting the complex and dynamic operation of the

distribution system. We have developed monthly (Fig. 8) and six-month averaged (Fig. 9)

contribution from the different sources at the distribution sites based upon 1-3 samples collected

each month; however, such maps can be developed at varying spatiotemporal scales depending

upon the purpose and application of the method.

To validate the results obtained by BIMM we compared the volumetric contribution of the

sources with their areal contribution. Volumetric and areal contributions were strongly and

systematically correlated across all sources (Table 1). However, our model predicted that the

Wasatch creeks supplied a larger fraction of the area than suggested by their volumetric

contribution, and that the Provo River sources supplied a smaller area than implied by volumetric

production numbers in May and June (Table 1). This discrepancy could reflect differences in



water demand across the service area, although most of the area our analysis suggests was served by the Wasatch creeks source is heavily populated, the corridor served by the Provo River water source includes more industrial development that may over-consume water per unit area.

Nonetheless, the overall similarity between the areal coverage estimated here and reported volumetric production numbers provides an additional line of evidence supporting the robustness of the BIMM.

**3.5 Model improvements and future application of BIMM in other urban water systems**

We have shown here that BIMM provides robust estimates of the contribution from different

sources to distribution sites within a PWSS. In our analysis, the isotopic compositions of major sources were distinct, allowing our model to quantify the contribution from the major sources at the distribution sites with robust estimates of uncertainty across the supply system. However, the robustness of our analysis was limited by non-unique solutions arising from distribution sites with isotope values intermediate to candidate sources. These challenges and limitations could be

addressed with the inclusion of other conservative tracers such as chloride, calcium, and strontium (and its stable isotopes) that might vary significantly between the different sources, thus providing additional constraints and improved model predictions. Additional system data, such as pressure and elevation gradients and flow velocity within the system, might also be included within the model to improve accuracy.

A key prerequisite for future successful implementation of the BIMM in other PWSSs is that all sources in the PWSS be characterized and have significantly distinct isotopic and/or geochemical signatures. In PWSSs with negligible isotopic and geochemical variability between the sources, the capacity of the BIMM to characterize the system would likely be limited. Finally, the BIMM approach is sample-based and an appropriate sampling design would be required to accurately

connect sources and distribution sites and extract meaningful information from the analysis. The

sampling design should consider factors such as source compositions, system operations, water

residence time, water demand, population density, etc., within the PWSS to develop a robust

sampling strategy for implementing the BIMM. It is essential to capture temporal variations,

especially for surface water sources or other sources with rapid water transit time, to establish

accurate association between the sources and distribution sites. In our analysis, our monthly

sampling protocol captured the successive isotope enrichment of the Provo River source that was

vital to the success of our model.

The framework applied here can be useful in establishing source water footprints, pathways, and

interactions of water sources within PWSS.  In cities across the developed world that use

hydrodynamic models (such as WaterCAD and EPANET) to predict water quality and

contaminant concentration across their supply systems, the accuracy of these predictions can be

evaluated by comparing the observed and predicted SIW (or other conservative geochemical

tracers) at several distribution sites using the hydrodynamic model. In many developing and

rapidly growing cities across the world where applying hydrodynamic models are challenging

and difficult, a framework similar to shown here, can be used to develop GIS products such as 1)

service maps of the different sources, 2) regions within the PWSS undergoing seasonal source

switching and 3) regions serviced by surface or groundwater respectively. These products can be

helpful in moderating water rights issues, tracking of source- and WTP-related contaminants,

evaluating the susceptibility to climatic variations and investigating long- and short-term effects

of source water quality on public health.




## 4. Conclusions

Water isotopes have been used extensively to monitor and understand the natural component of water cycle (*Gat*, 1996; *Aggarwal et al.*, 2005; *Bowen and Good*, 2015), however their
application in urban water systems has been limited. Recent work has shown the capacity of water isotopes to record information about water management and quantify effects of climatic variability on water resources (*Jameel et al.*, 2016; *Tipple et al.*, 2017). Moving beyond the coarse resolution of these studies, our work has highlighted the ability of water isotopes to provide information about PWSS operation at a much finer scale. Here, we have shown the
ability of water isotopes to provide estimates of the contributions of multiple water sources across a large metropolitan PWSS and inform our understanding of the physical structure and operation of the system. The method developed here does not rely on independent information about pipe networks, flow velocities, pressure gradients or other details of the PWSS that are integral to hydrodynamic models, and thus can be used to interrogate PWSS where this
information is lacking or to independently validate hydrodynamic model results. Our application used only two isotope ($\delta^2$H and $\delta^{18}$O) measurements, supplemented with information on source volumes and geographic locations. Future applications could improve upon our work by including additional geochemical tracers, flow rates, adding additional information on distribution system structure (where available), collecting samples with higher spatiotemporal
resolution and refining the statistical model. Considering that stable isotope analysis of most water samples is now rapid (minutes) and inexpensive, geochemically-based BIMMs offer an attractive tool for studying and monitoring PWSS in support of management and water security.



## Acknowledgements

We thank JVD for collaborating on this project and Heidi Nilsson for collecting the water

samples. This work was supported by U.S. National Science Foundation grant 1208732.

## Code and data availability

Raw data sets analyzed within this study can be accessed through Waterisotopes Database

(http://waterisotopes.org, ProjectID = 00065) and are also available on Hydroshare

(https://www.hydroshare.org/). An R code to execute equations described in section 2.4 for the

month of June 2015 is provided as supplementary material, as are the source and distribution site

dataset for June 2015.





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





Table 1: Comparison between volumetric (V) and areal (A) contributions of the different sources from May 2015 to October 2015. JVTP and SETP are considered as separate sources in May and June 2015 and combined sources from July to October 2015. Sources contributing less than 1% of the total volume have been grouped together as "Minor sources" for June, July and August 2015. All values are in percent.

| May 2015 | | | June 2015 | | | July 2015 | | |
|---|---|---|---|---|---|---|---|---|
| Source | V | A | Source | V | A | Source | V | A |
| | | | | | | | | |
| Provo River (JVTP) | 60.6 | 47.7 | Provo River (JVTP) | 80.5 | 63.9 | Provo River (JVTP and SETP) | 86.2 | 87.1 |
| Wasatch Creeks (SETP) | 24.3 | 30.2 | Wasatch Creeks (SETP) | 9.2 | 27.3 | SWTP | 2.6 | 1.7 |
| SWTP | 6.5 | 5.7 | SWTP | 3.8 | 2.7 | 64S/70S well | 2.8 | 2.9 |
| Solena Way well | 4.6 | 6.4 | 64S/70S well | 2.4 | 2.4 | Siesta well | 1.9 | 1.8 |
| 64S well | 3.0 | 7.1 | Solena Way well | 1.0 | 0.8 | 18E well | 1.6 | 1.4 |
| 45 S well | 1.0 | 2.8 | Minor sources | 3.1 | 2.9 | Monitor well | 1.5 | 1.6 |
| | | | | | | Minor sources | 3.3 | 3.2 |

| August 2015 | | | September 2015 | | | October 2015 | | |
|---|---|---|---|---|---|---|---|---|
| Source | V | A | Source | V | A | Source | V | A |
| | | | | | | | | |
| Provo River (JVTP and SETP) | 86.2 | 86.9 | Provo River (JVTP and SETP) | 92.1 | 91.3 | Provo River (JVTP and SETP) | 90.0 | 94.3 |
| SWTP | 5.4 | 4.5 | SWTP | 2.7 | 2.3 | SWTP | 6.1 | 1.7 |
| 64S/70S well | 3.9 | 2.8 | 64S/70S well | 2.8 | 3.4 | 64S/70S well | 3.2 | 3.1 |
| 90S well | 2.3 | 2.2 | 90S well | 1.4 | 1.9 | 90S Quail well | 0.7 | 0.8 |
| Minor Sources | 2.2 | 3.4 | 90S Quail well | 0.9 | 1.1 | | | |







Figure 1: Jordan Valley Water Conservancy District (JVD) wholesale area (white) and Jordan

Valley retail area (purple) within the Salt Lake metropolitan valley (black border). The aqueducts

from Provo River and the Wasatch Creeks are shown for illustrative purposes only. Source of

base map: ESRI digital media.




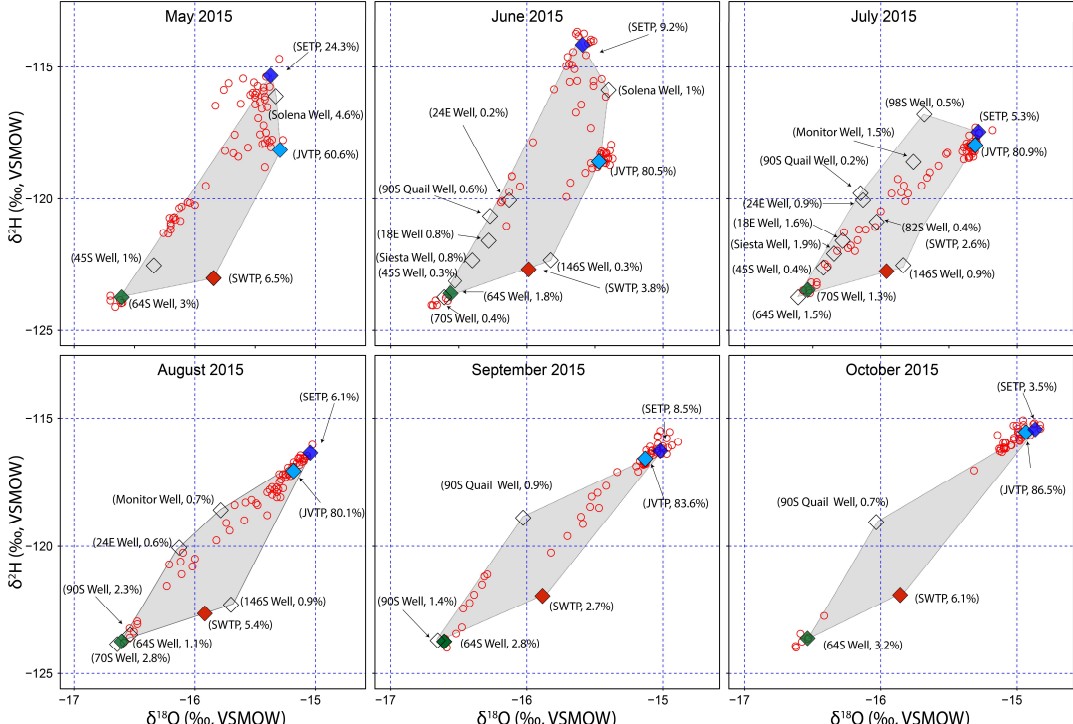

Figure 2: Sources and distribution site isotope ratios from May 2015 to October 2015. Red

hollow circles and diamonds represent distribution sites and sources respectively. The four major

sources (JVTP, SETP, SWTP and 64S well) have been colored light blue, dark blue, orange and

green respectively. The grey region is the convex hull of the sources (defined as the minimum

area enclosing all the source isotope values).





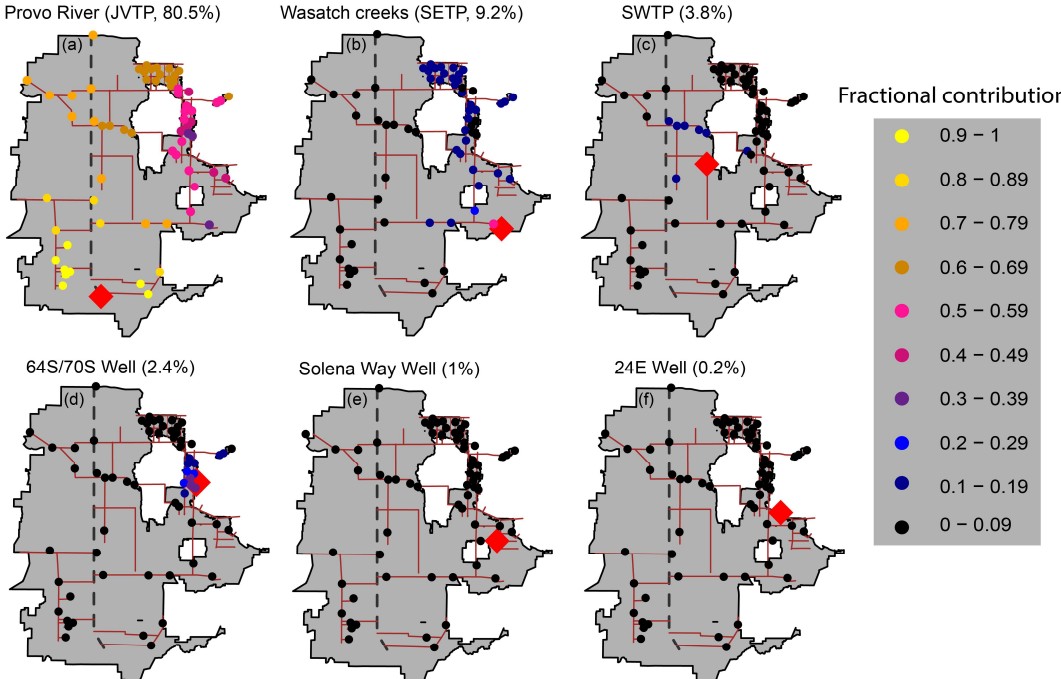

Figure 3: Mean prior contribution of selected sources at the distribution sites for June 2015 based
upon Eq.6 described in section 2.4. Distribution sites are shown as circles, and the color reflects
the assigned prior contribution from the different sources. The source location is shown as red
diamond in each panel. The name of each source and its percent volumetric contribution is
shown above each panel.





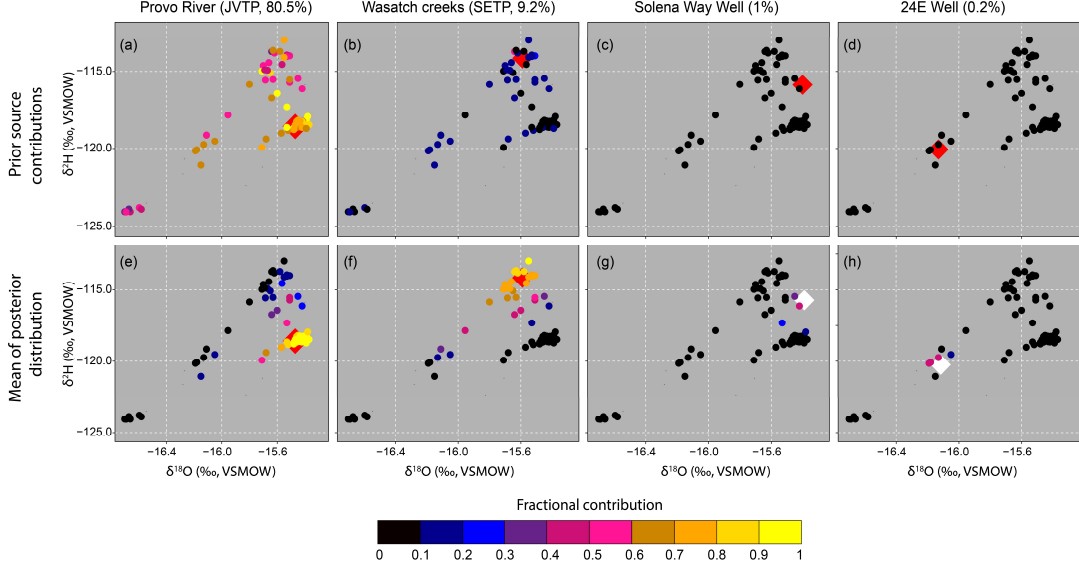

Figure 4: Prior contribution of selected sources at the distribution sites (a-d) and mean posterior

contribution of selected sources at distribution sites (e-h) in isotope space for June 2015. Red

diamonds represent sources and the circles represent distribution sites. For clarity, diamonds in

panels A and B have been enlarged and in panel B3 and B4 are shown in white.





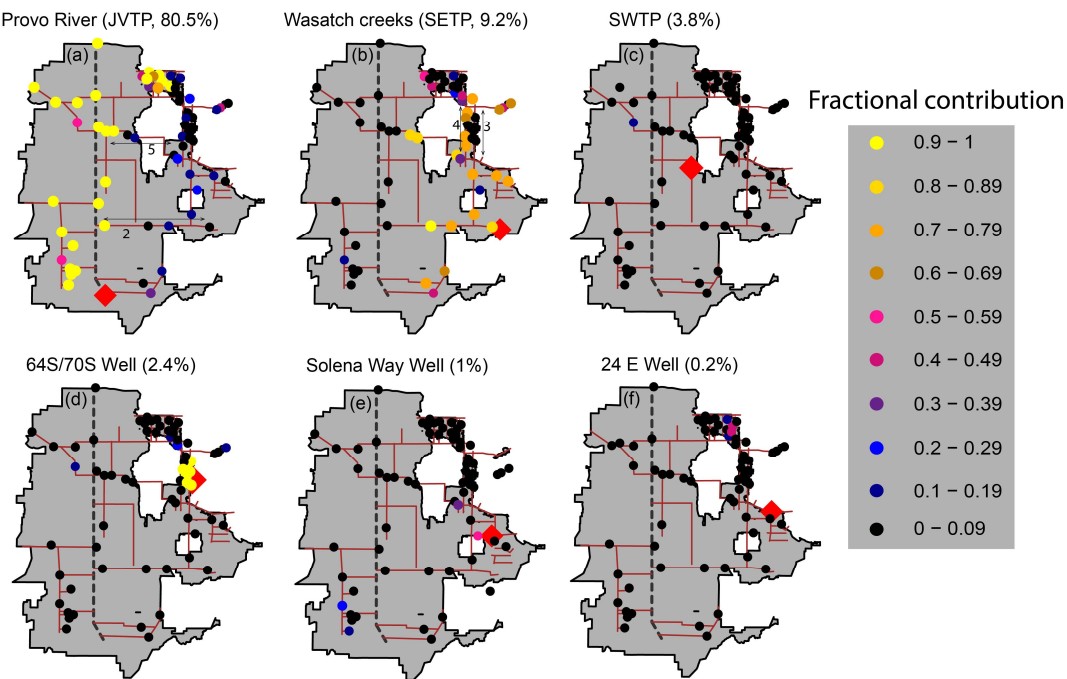

Figure 5: Mean of the posterior contribution of selected sources at the distribution sites for June

2015. Distribution sites are shown as circles and the color reflects the mean of the posterior

contribution from the respective source at that site. The source in each panel is shown as a red

diamond. Name of the source and its percent volumetric contribution is shown above each panel.

Transmission lines 2 and 5 are shown in panel (a) and lines 3 and 4 are shown in panel (b).





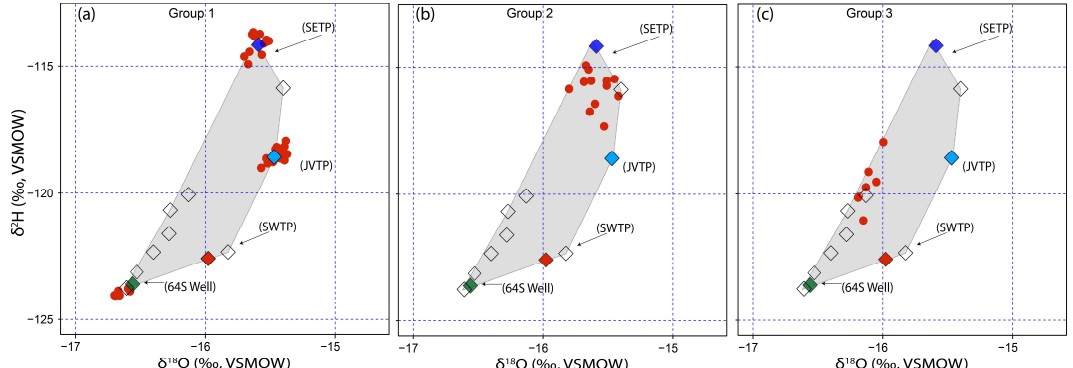

Figure 6: Distribution sites and sources for June 2015 shown as red circles and diamonds. The

four major sources (JVTP, SETP, SWTP and 64S well) have been colored light blue, dark blue,

orange and green respectively and are labelled. Minor sources are shown as hollow diamonds. (a)

group 1, (b) group 2 and (c) group 3 distribution sites.





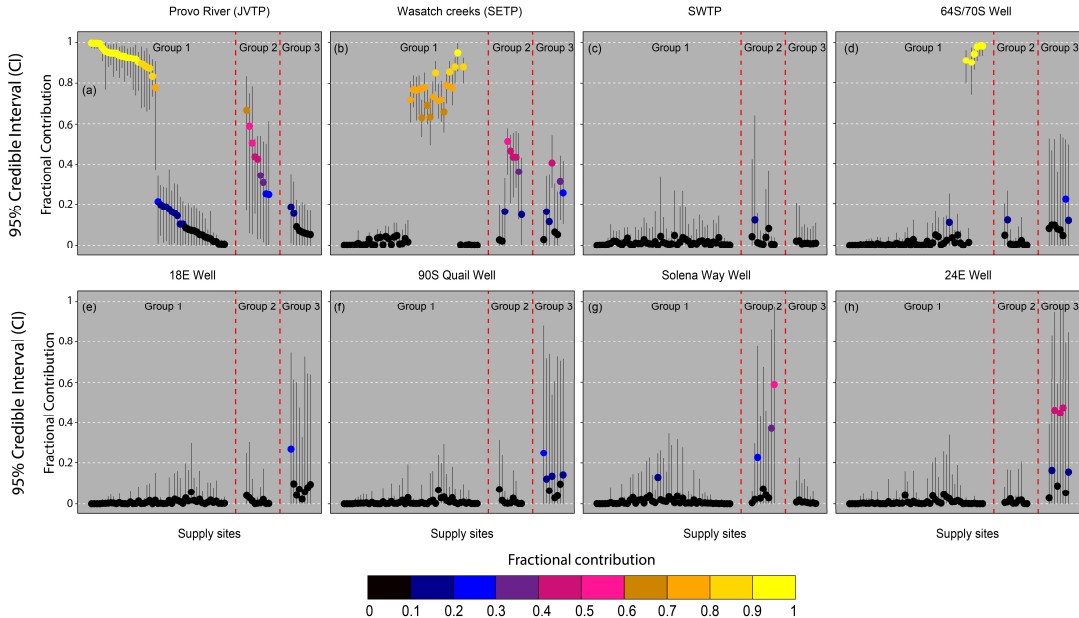

Figure 7: Mean (circle) and 95% credible interval (vertical black lines) associated with the source contributions at the distribution sites for the different groups. Sites in panel a have been sorted with decreasing contribution from JVTP. The same sorting order is maintained for all the panels (b-h). Red diamonds represent source and the circle represent distribution sites.





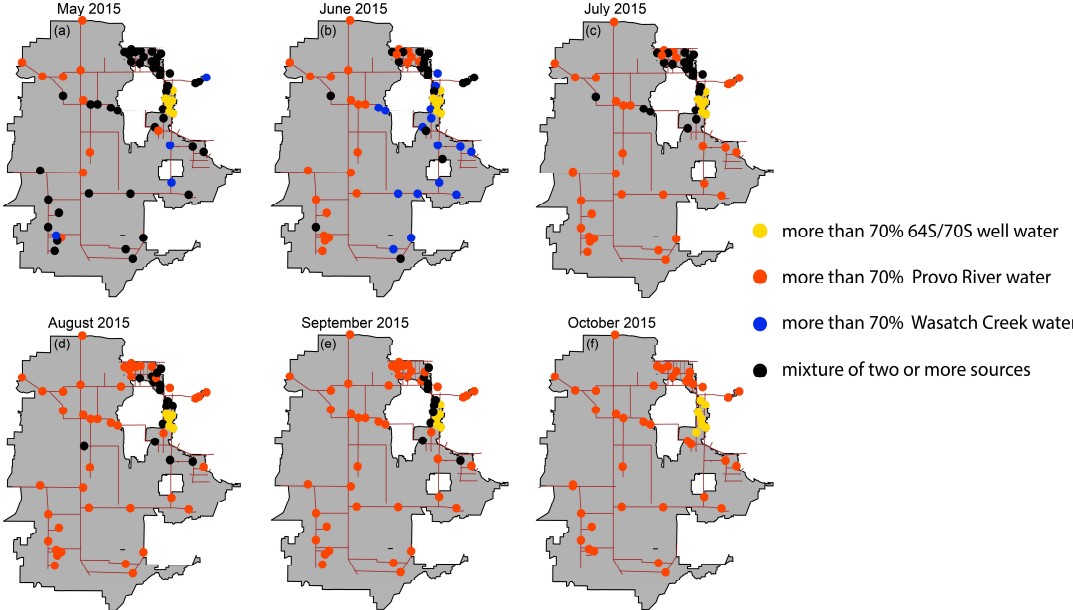

Figure 8: Spatiotemporal variation in sources and distribution sites connectivity from May 2015 to October 2015. Distribution sites receiving more than 70% water from a single source are shown in orange, blue and yellow circles and sites receiving water from multiple sources (less than 70% water from a single source) are shown in black circle.



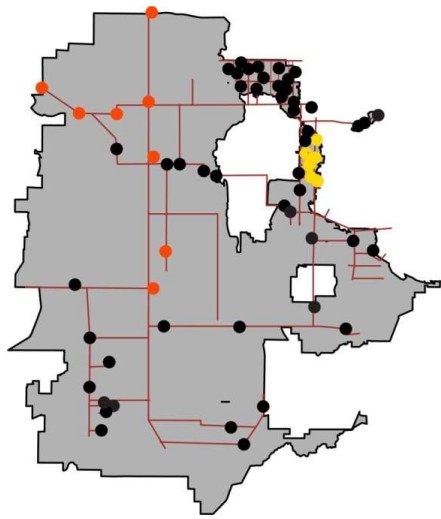

Figure 9: Mean contribution of different sources at the distribution sites during May 2015 to

October 2015. Sites in orange and yellow circles received water primarily (>70%) from Provo

River and 64S/70S well, respectively, throughout the entire sampling period. Sites in black

circles received water from multiple sources or switched sources at least once during the

sampling period.