# Peer review of "Isotopic reconnaissance of urban water supply system dynamics"

_Hydrology and Earth System Sciences, 2018_

## Referee Comment (RC1) · R. van Geldern (Referee) · 8 Aug 2018

**Manuscript ID: HESS-2018-283**

*„Isotopic reconnaissance of urban water supply system dynamics"*
Jameel et al.

**Comments to the authors**

    *General remarks*
The manuscript describes the attempt to identify source contributions to the Salt Lake Valley (USA) to the local public water supply systems (PWSS) by stable isotope analysis rather than using the hydrodynamic models provided by the water suppliers. As a European, I am still surprised that the water suppliers in the US obviously have no exact data about their contribution of the various water sources to tap water and what was delivered to each household. But I notice that this study serves as an example for developing countries where such data is definitely not readily available.

The manuscript is adequately organized and contains no grammatical or orthography errors. Text is concise and to the point. The English is fine and does not need any revision; note that I am not a native speaker.

Overall, I would recommend this study for publication with some minor revisions outlined in more detail below.

    *Specific comments*
L95-100    Publications from the authors themselves (and others) exist of the topic of SIW in urban systems. This usage is not restricted to the validation of hydrodynamic models but has also been used to detect for instance sewer infiltration. A few studies on urban system and SIW exist and I think this is the place in this study to cite them? It is not the case that SIW are totally new to urban systems (this sentence reads a bit like that) and the authors should be aware of that as they already cited Jameel et al (2016) above.

De Bénédittis, J. and Bertrand-Krajewski, J. L. 2005. Measurement of infiltration rats in urban sewer systems by use of oxygen isotopes. *Water Sci. Technol.* **52**, 229-237.

De Bondt, K., Seveno, F., Petrucci, G., Rodriguez, F., Joannis, C. and Claeys, P. (2018): Potential and limits of stable isotopes ($\delta 18O$ and $\delta D$) to detect parasitic water in sewers of oceanic climate cities. - *Journal of Hydrology: Regional Studies*, **18**, 119-142.

Ehleringer, J. R., Barnette, J. E., Jameel, Y., Tipple, B. J. and Bowen, G. J. 2016. Urban water – a new frontier in isotope hydrology. *Isot. Environ. Health Stud.* **52**, 477-486.

Jameel, Y., S. Brewer, S. P. Good, B. J. Tipple, J. R. Ehleringer, and G. J. Bowen (2016), Tap water isotope ratios reflect urban water system structure and dynamics across a semiarid metropolitan area, Water Resour. Res., 52(8), 5891-5910.

Kracht, O., Gresch, M. and Gujer, W. 2007. A Stable Isotope Approach for the Quantification of Sewer Infiltration. *Environ. Sci. Technol.* **41**, 5839-5845.

L101-119    This paragraph reads a bit like an abstract or a brief version of objectives, methods, results and conclusions. I suggest it is enough here to state where the study occurred and the respective objectives. This is also the place to cite the other studies (that I also reviewed) that already explored the Salt Lake Valley (SLV) metropolitan area by stable isotopes (Ehleringer et al. 2016, Jameel et al., 2016). Please outline here briefly which questions were not asked and/or could not be answered be these earlier studies that already

used SIW in the SLV area and how this relates/led than to this study. Here the reader get's the impression that SIW are applied for the very fist time to the study area, which is not the case.

L138        Switch units. Please refer to SI units first (cubic meters or liters); you can then add imperial units (gallons) in parentheses if necessary.

L136-150        It is of interest what kind of treatment is necessary before it can be sold/distributed. The various processes involved might be very basic or technically challenging (i.e. from simple sand bed filtering to ultrafiltration and/or chemical treatment). Some of these processes might also influence the stable isotope ratio because of secondary processes such as evaporation or mixing. In addition, this might change over the year and the specific treatment location and therefore add additional bias to your data. Any ideas how to deal with that?

L176        This is unclear to me. You measured the isotope ratio of some wells for a specific(?) month and then used this value also for all other months of this study? Or the other way around: you used isotope values measured outside the sampling interval of this study and used them here? Please clarify.

L186        Use *italic* characters for the delta symbol (Coplen 2011; Brand etal., 2014) throughout the text.

Coplen, T.B. (2011): Guidelines and recommended terms for expression of stable-isotope-ratio and gas-ratio measurement results. - *Rapid Communications in Mass Spectrometry*, **25**, 2538-2560, [doi:10.1002/rcm.5129].

Brand, W., A., Coplen, T., B., Vogl, J., Rosner, M. and Prohaska, T. (2014): Assessment of international reference materials for isotope-ratio analysis (IUPAC Technical Report). - *Pure and Applied Chemistry*, **86**, 263-467, [doi:10.1515/pac-2013-1023].

L192        'van' missing in citation: "(van Geldern and Barth, 2012)" => place under "V" in reference list (not "G")

L295        Not sure if the term 'deuterium excess' has so far been introduced. If not you cannot use "d-excess" without giving the full name here. And: please use either 'deuterium excess' or *d* (but not 'd-excess').
Further: I do not think that a *d*-value of "~10‰ is high". This is the normal expected *d*-value from the GMWL? Or do you mean that the d-values from well 64S are higher compared to your other sources? Please clarify.

L300/301        round *d* to one decimal or less (5.1‰ or 5‰ but not 5.19‰)

L471-482   I think this is an important point and this cross-check was announced earlier in the manuscript (section 2.5). I suggest giving this section a more prominent position either earlier in the discussion and/or upgrade to a sub-chapter in section 3.

Further: From my first impression of Table 1 the numbers are in quite good agreement although some discrepancies between the BIMM (this give V, right?) and the areal contribution (A) exist for some sources. This raises the question if the stable isotope method is really necessary to identify sources if it can be done rather easy without measuring isotopes in many cases? The authors should more highlight the differences rather than the agreement between the two approaches. If both approaches come to comparable results, then I cannot see a reason running SIW?

L486        'Distinct sources' (in terms of their isotope values) are a very critical point. If various sources in a system are NOT distinct (enough) the isotope approach will give no unique solution or will simply fail. I notice this is mentioned again in lines 496-498. Please include here a statement that the BIMM is not only 'limited' but will not work (or gives no useful solution; it will probably calculate something…).

**Tables 1**

Table 1    Please clarify in the captions which of the values (A or V) is the result if the
BIMM (aka received from the isotope mixing model).

*Technical comments*

P4L88      space character after parentheses

L156       Fig. 1 (not fig. 1)

L164       delete 'as'

L184       *t*-test (*t* is italic)

L185       delete 'in' (before 'for')

L192/193   delta symbol italic; R italic

L196       space character between number and unit (±0.3 ‰); change throughout the text;
NIST style convention #15; see: <https://physics.nist.gov/cuu/Units/rules.html>

L2010      space character missing before parentheses

L231       space character missing '1 assigned'

L256       comma missing before 'respectively'?

L256       fig. => Fig.

---

## Referee Comment (RC2) · Anonymous Referee #2 · 9 Aug 2018

General comment This manuscript presents the application of an isotopic (2H and 18O) approach used to quantify the contribution of different water sources in a complex public water supply system in Utah, identifying areas supplied chiefly by one source and those supplied by multiple sources), as well reconstructing basic flow patterns. This research is highly relevant for practical purposes and has operational implications in urban and rural environments, and will surely be of interest to the readers of HESS.

The manuscript is very well written, logically organized and nicely illustrated. The dataset is well explored and analysed in a robust way, and the results and their interpretations are solidly supported by data. I have only some minor comments and clarifications requests for the Authors before recommending this manuscript for publication.

Specific comments The manuscript contains an excessive number of abbreviations (e.g., PWSS, SIW, SLV, BIMM, JVD, WTP, MGD, SEPT, SWTP etc) that sometimes make the reading difficult. Please, use acronyms sparingly, only when strictly necessary.

Introduction: although the results are clear, the Introduction misses a clear definition of the working hypothesis upon which to establish specific objectives. The objective should stem from the analysis of the identification of research gaps in the current literature and/or from a practical management issue in the study PWSS. Pease, re-structure the introduction taking this into consideration.

L8. Figure 2 is introduced quite abruptly in the text, before the M&M section, even though it seems to me to present some results of this study. I suggest redefining its position in the structure of the manuscript.

In the M&M section, please specify in a clearer way the concept of prior and posterior fractional contributions (see also captions of Fig. 3, 4 and 5).

L208. These assumptions should be discussed.

L290. The Hotteling multivariate t-test assumes a multivariate normal distribution: is the same distribution assumed at L209?

L296-301. I suggest reporting d-excess values as indicators of evaporative effect, to corroborate these statements.

L314. Please, be more specific: in what sense this work goes beyond previous work? This is part of the discussion on the originality and novelty of this research.

L419. I suggest moving this paragraph to section 3.5.

Minor comments and technical corrections

L45. Please, specify what these particular threats are.

L91. The reference to Gorski et al., 2015 applies to vapour. Is still relevant?

L122. Please, specify that the study was carried out in the USA for the benefit of the international readers.

L129. Is this especially true for irrigation purposes?

L135. Why? To avoid overexploitation? Please, specify.

L138. Please use SI units.

L146. Why do these well have so high salt concentrations?

L163: Combine with L174.

L184. Specify the sample size.

L199. The acronym has been already defined at L115

L201. Information, such as?

L202-204. Please, clarify.

L345. Typo: to our model.

L355. Typo: lines.

L380. Can you explain why you used credible intervals and not confidence intervals?

L396. Replace "credible intervals" with "CI".

L510. Give references for these two models.

L513. I doubt cities in developing countries have to fund to perform costly isotope analysis over a long time span or large areas. Perhaps add a comment here.

Fig. 1. In the caption define WTP and SWTP.

Fig. 7, L780. Replace panel a with (a).

[Figure]

---

## Author Comment (AC1) · 22 Sep 2018

Dear Dr. van Geldern,

Thank you for reviewing our manuscript. Please find attached our response to your comments and also the updated manuscript (reply_reviewer_1.zip).

Please also note the supplement to this comment:
https://www.hydrol-earth-syst-sci-discuss.net/hess-2018-283/hess-2018-283-AC1-supplement.zip
* * *

---

## Author Comment (AC2) · 22 Sep 2018

Thank you for reviewing our manuscript. Please find attached our responses to your comments and also the updated manuscript (reply_reviewer_2.zip).

Please also note the supplement to this comment:
https://www.hydrol-earth-syst-sci-discuss.net/hess-2018-283/hess-2018-283-AC2-supplement.zip